# Risk Management: Rethinking Fashion Supply Chain Management for Multinational Corporations in Light of the COVID-19 Outbreak

**May McMaster [1], Charlie Nettleton [1], Christeen Tom [1], Belanda Xu [1], Cheng Cao [1] and Ping Qiao [2],***

1   Business School, University of Sydney, Sydney, NSW 2006, Australia; mmcm7306@uni.sydney.edu.au (M.M.); cnet3239@uni.sydney.edu.au (C.N.); ctom8704@uni.sydney.edu.au (C.T.); bexu4940@uni.sydney.edu.au (B.X.); ccao9642@uni.sydney.edu.au (C.C.)
2   School of Management and Economics, Beijing Institute of Technology, Beijing 100081, China
*   Correspondence: ping.q.0306@gmail.com

**Abstract:** Through an international business risk management lens, the widespread and catalytic implications of the 2020 COVID-19 pandemic on the supply chains (SCs) of fashion multinational corporations (MNC) are analyzed to contribute to existing research on supply chain management (SCM). While a movement towards agile, networked supply chain models had been in consideration for many firms prior to the outbreak, the pandemic highlights issues inherent in supply chains that employ concentrated production. We examined the current state of fashion supply chains, risks that have arisen historically and recently, and existing risk mitigation methods. We found that while lean supply chain management is primarily favored for its cost and waste reduction advantages, the structure is limited by the lack of supply chain transparency that results as well as the increasing demand volatility observed even before the COVID-19 outbreak. Although this problem might exist in the agile supply chain, agile supply chains combat this by focusing on enhancing communication and buyer-supplier relationships to improve information exchange. However, this structure also entails an associated increase in inventory and inventory costs. The COVID-19 pandemic has caused supply and demand disruptions which have resonating effects on supply chain activities and management, indicating a need to build flexibility to mitigate epidemic and demand risks. To address this, several strategies that firms can adopt to control for such risks are outlined and key areas for further research are identified which consider parties both upstream and downstream of the fashion supply chain.

**Keywords:** COVID-19; risk management; supply chain risks; fashion industry; supply chain management

## 1. Introduction

Following World War II and the industrialization of post-war economies, rapid globalization saw firms extend fashion production and manufacturing to developing markets in order to capture cost efficiencies through outsourcing, alliances, and foreign direct investment (FDI). In the mass-market segment, as consumer demand for choice and low-cost increased, competition intensified and profit margins decreased, resulting in the movement of supply chain (SC) activities from countries with high labor costs to those with lower costs in Asia, Europe, and Africa (Graafland 2002). While cost efficiency is often viewed as the key motivation for large corporations to undertake international outsourcing (Di Gregorio et al. 2009), other drivers include market expansion, talent-sourcing, proximity to points of sale, and material-seeking (Caniato et al. 2015).

Since the adoption of global SCs, the fashion industry can be considered in terms of its volatility, velocity, variety, complexity, and dynamism (Čiarnienė and Vienažindienė 2014; Mustafid and Jie 2018), and SC in the fashion sector is full of uncertainty and unpredictability (Giannakis and Louis 2016; Mustafid and Jie 2018). Fashion multinational corporations (MNC) have been subject to key risks that have often been realized, such as late deliveries, long lead times between returns and resending to customers, stock-out or over-stock, and deliveries in single solutions (Martino et al. 2015). Moreover, the internationalization of SCs has exposed fashion MNCs to country-specific risks, which leads local high-risk events to snowball in magnitude (Ivanov 2018; Bevilacqua et al. 2019a). Due to the COVID-19 outbreak, in the midst of the pandemic with countries varying in their methods and capacity to contain the virus, social distancing and lock-down requirements limiting production in key supplier markets reveal the shortcomings of existing supply chain management (SCM) models.

With the COVID-19 pandemic posed to catalyze a movement towards more agile SCs, our discussion of the merits and limitations of lean compared to flexible SCM will contribute to existing literature, which fails to give meaningful consideration to the long-term structural changes that will prevail following the COVID-19 pandemic. Section 2 holistically reviews and expands on SC risk management issues in light of the COVID-19 outbreak and an overview of theory underpinning modern SCM. In Section 3, we discuss the merits and faults of lean and agile SCM styles before analyzing the risks highlighted and drawn out in the COVID-19 period. Based on background and theoretical review, Section 4 is the implications of COVID-19 for SC risk management. We discuss frontier issues in Section 5, while the conclusion is shown in Section 6.

## 2. Background and Theoretical Foundation

### 2.1. COVID-19 Background

The highly-infectious COVID-19 virus was declared a global pandemic by the World Health Organization on 11 March 2020 (Armani et al. 2020). Although its exact origins are unknown, the COVID-19 pandemic is believed to have emerged in Wuhan, China in December 2019. The severity of the virus differs between individuals, ranging from mild symptoms of fever, coughing, and shortness of breath to severe respiratory problems in critical cases. A notable number of cases have resulted in hospitalization and even death (Zhou et al. 2020b). On 3 March, it was estimated that the global mortality rate of COVID-19 was approximately 3.4% (WHO World Health Organization). COVID-19 mortality has been more common in older adults and those with pre-existing health conditions (Zhou et al. 2020a). The operations of many organizations have been severely disrupted as the outbreak spread around the globe, impacting both supply and demand (Ivanov 2020). The unprecedented nature of the pandemic has meant that businesses had no prior planning and were exposed to significant risk. A survey conducted by Ernst & Young in 2019 found that of 500 senior board members globally, only 20% of the executives were confident that their companies were prepared to respond to a large adverse risk (Ernst and Young 2020). While most short- or medium-term impacts of COVID-19 can now be identified, the long-term impacts still remain uncertain. The pandemic has broken many global SCs (Araz et al. 2020), particularly for organizations with lean and globalized SC structures. In fact, it was reported that 94% of the Fortune 1000 companies have experienced COVID-19-driven SC disruptions (Sherman 2020).

As a result, organizations have been pushed to undergo significant work to re-design SCs, improve resilience, and reexamine relationships with suppliers in order to reduce systemic risks.

### 2.2. Supply Chains in the Fashion Industry

The components and nature of SCs in the fashion industry differ between companies and are dependent on factors including their products, target market, competitive priorities, global strategy, lead time, delivery delays, and supplier integration (Kim 2013; Fisher 1997; Lee 2002; Mehrjoo and Pasek 2016; Li et al. 2016). SC literature has produced a large body of work analyzing a variety of approaches to SCM

for fashion companies. The fashion apparel industry, and fast fashion in particular, has been characterized by short life cycles, high demand volatility, low predictability, and high impulse purchasing (Mehrjoo and Pasek 2016; Martino et al. 2015). As components of SCs vary across the fashion industry, so do SCM strategies. The effectiveness of various SCM strategies in improving efficiency and mitigating risk has been explored across a wide range of SC literature.

Kim (2013) conducted a series of interviews with managers of large companies to determine factors that shape the SCM strategy. It was concluded that there is a strong link between the competitive priorities of a company and their choice of target market, which determines a company's choice of SC strategy. One of the most drawn upon studies in SCM is from Fisher (1997), who suggested that the optimal SCM strategy is dependent on the nature of a firm's products. The study highlights two product categories: functional products, which should be coupled with a physically efficient SC; and innovative products, which should be coupled with a market responsive strategy. Alternatively, Lee (2002) categorizes products according to their reliability of demand. Products with stable demand and products with highly volatile demand should not be managed in the same way. When product demand is unpredictable, there should be a strong focus on matching strategy with the uncertainties in demand and supply.

It is typical for companies in the fashion industry to be faced with the dual pressure of maintaining short lead times and low costs (Shen and Chen 2019). This has resulted in the formation of buyer-driven and geographically complex outsourced SCs. The effect of lead time on SC performance (inventory, cost, backlog, and risk) is the key to success for fast fashion sectors (Mehrjoo and Pasek 2016). To address these pressures, Shen and Chen (2019) suggest that a successful SCM strategy should be focused on developing relationships with the outsourced fashion SC. Similarly, Martino et al. (2015) and mboxciteauthorB55-jrfm-861287 (2016) argued that supplier integration can significantly promote financial performance and help to mitigate the negative effect of a financial tsunami on the financial performance of the fashion enterprises. Although this literature on SCM emphasizes the importance of supplier relationships, it is limited in that it does not consider the importance of a flexible SC. Relying on a small number of suppliers can heighten risk of SC disruption in unexpected circumstances, as seen with the global COVID-19 pandemic. In the fashion industry, however, there has been a trend to minimize operating costs and companies often forwent flexibility in order to achieve a lean SC. The main challenges in fashion SCs are the high demand volatility of customers (Choi 2006, 2007), the dynamic market, and market uncertainty (Gligor et al. 2015; Mustafid and Jie 2018), which can create problems of either overstocking or understocking (Choi 2006, 2007). Here, the importance of an agile SC is observed.

## 2.3. Supply Chain Disruption Risk and Mitigation Strategies

The primary factors behind SC disruption and the ways for companies to best mitigate the associated risks have been explored in recent SC literature. First, the decision of whether to adopt a complex SC structure versus a simple SC structure has been debated. Ivanov and Ivanov and Dolgui (2019) argue for more simplistic SCs and emphasize that complex networks are likely to be more vulnerable to severe disruptions, where a hold up in one component of the SC can impact the subsequent components in the value chain. The impact of unexpected events and the extent to which they can influence SCs have also been explored. Ivanov (2020) uses simulation-based analysis to research the impacts of COVID-19 on global SCs, highlighting its impact on factors essential to SC performance including the timing of closing and opening of the facilities at various degrees, lead time, speed of epidemic propagation, and the upstream and downstream disruption durations.

In terms of mitigating SC disruption risk, studies have suggested that flexibility and diversification is the best way to hedge this risk. By researching the domino effect of factors affecting supply chain resilience (SCR) in the fashion industry, Bevilacqua et al. (2019b) suggest that due to manufacturers highlighting flexibility in order fulfillment, a flexible production structure is vital to effectively address

unpredictable turnarounds of the market in a timely manner. Likewise, in the research note about COVID-19 and SCR, Ivanov and Das (2020) indicate that the focus of SC resilience management should shift towards situational responses to real-time changes. In cases of unlikely but severe disruptions to SCs, temporary sourcing diversification could prove to be an effective response strategy. Finally, Sreedevi and Saranga (2017) discuss how supply, manufacturing, and distribution flexibility moderate the relationship between environmental uncertainty and supply risk.

In regards to the appropriate SC structure, which implies SC structure adaptation and flexibility, severe disruptions can change the SC structure and are involved with SC structural dynamics (Ivanov and Dolgui 2019, 2020). A study by Bevilacqua et al. (2019b) proposed a method that assists companies to better evaluate the hidden chain reaction and impact of a unique trigger event and predict how severely it would disrupt the SC. This type of modeling is useful for companies making decisions on a SC structure. Motivated by COVID-19 outbreak, Ivanov and Dolgui (2020) propose the integrity of the intertwined supply network (ISN) and viability. An ISN is an entirety of interconnected SCs that secure the supply of society and markets with goods and services, while SC viability management would instead be altered towards the situational reactions (Ivanov and Das 2020; Ivanov and Dolgui 2020).

Issues experienced by SCM are the *bullwhip effect* and *ripple effect*. The former reveals distorted information between entities within the supply chain, increasing the inventory holding and management costs and increasing lead time (Snyder and Shen 2019), while the latter occurs when a disruption triggers a chain effect downstream and upstream which influences SC performance (Bevilacqua et al. 2019b). The issue of the *bullwhip effect* is compounded in scenarios where the demand patterns are often uncertain and not effectively communicated from retail end to the supplier ends. Sarkar and Kumar (2015) explore the risk mitigation strategy of sharing real-time information of SC risks between manufacturers and retailers. This strategy minimizes the impact of the disruption by allowing inventory decisions to be adjusted to avoid overstocking or understocking, reducing the *bullwhip effect* and its associated costs. To reduce the *ripple effect*, Ivanov et al. (2019) investigate the role of digital technologies and Industry 4.0 in rising demand responsiveness and capability flexibility. With the support of big data analytics (BDA) and tracking and tracing system (T&T) technologies, Industry 4.0 increases the ability to reconfigure resources at the recovery stage (Ivanov et al. 2019).

The reviewed literature on SCM provides an insight into the factors that contribute to choices of SC structures. The question as to which SC structure is most effective in mitigating SC disruption risk has been widely debated and leaves room for further discussion on the benefits of a lean SC structure versus pursuing a more agile, diversified approach.

## 3. Supply Chain Management Theories and Applications

There are four common categories of SC strategies that firms implement to improve their operational efficiency given the market they are serving: (1) high cost efficiency, (2) risk hedging by pooling and sharing resources to reduce disruption impacts, (3) high responsiveness and flexibility to change, and (4) agility that entails a rapid and proactive response to volatility within the market while reducing supply disruptions (Lee 2002). Three strategies that are specific to the fashion industry are speed, cost advantage, and brand equity (Mehrjoo and Pasek 2016). Although all three strategies are desirable for all fashion segments, luxury brands often focus on brand equity where emphasis is given to style, design, and superior quality of material and build. To better manage the product quality, these brands employ a vertically integrated SC with most, if not all, manufacturing and sourcing completed in-house (Robinson and Hsieh 2016). While numerous SCM structures exist, we focus on comparing lean SCs to agile SCs, as they are the most practiced SCMs within the industry.

### 3.1. Lean Supply Chain Management

Historically, managing suppliers and manufacturers have been difficult due to scarcity of resources and cost-efficient labor. This scenario gave an imperative for firms to execute lean SCM, where optimized

inventory management is promoted. As a strategy, lean SCs focus on cost minimization and waste elimination. Wu and Wee (2009) explain the lean SC strategy as the reduction of non-value-added operations and improvement of value-added processes, where the leanness is applied upstream (Mason-Jones et al. 2000). Lean SCs are also associated with *zero inventory management* (Fan et al. 2007) and *Just-in-Time* (Wu 2009), a method developed by Toyota to reduce lead times in production by optimizing inventory management. Although optimized inventory management entails reduced inventory costs, responsiveness to urgent deliveries and order fulfillment rates are paramount for success using this strategy (Cabral et al. 2012). Reduced inventory often portends reduced resilience, and this trait is often undesirable in the fashion industry, where demand is often volatile and seasonality dictates trends.

　　Within the retail fashion industry, one of the prominent beneficiaries of lean SCs has been firms within the fast fashion industry. Fast fashion grew dramatically in the 1990s by offshoring production to low-cost manufacturing countries such as Bangladesh, China, and India (Allwood et al. 2006) where lean supply strategies were executed to reduce wastage and improve cost efficiencies. Furthermore, the introduction of The Regional Comprehensive Economic Partnership (RCEP) between ten ASEAN members and six other large economies in the Asia-Pacific region including China, India, Japan, South Korea, and Australia, provided a highly integrated regional SC (Kim 2016; Lee 2016) where a combination of discriminatory tariff elimination (Baldwin and Wyplosz 2006) and high barriers for entry for non-RCEP members, provided a competitive advantage for suppliers within this area. The comparative advantage between the countries meant sharing resources and materials, and collaboration of specialized skilled labor and manufacturing facilities (Lopez-Acevedo and Robertson 2016). This provided fast fashion firms with further incentives as textiles and apparel manufacturing requires high labor intensity due to limited opportunities for automation (Lu and Dickerson 2012). Additionally, increasing inflexibility and complexity in the incumbent Asia-Pacific-based SC (PwC 2013; Gray et al. 2013; D'Arpizio et al. 2014), as well as a combination of diminishing labor cost advantages and increasing logistics and coordination costs (Bailey and Propris 2014; Fratocchi et al. 2014), compelled the brands to adopt lean SC strategies as a measure to decrease cost measures and improve profit margins. Companies such as Zara and H&M offshored their manufacturing and sourcing to these countries where they enjoyed low-cost labor as a competitive advantage (Allwood et al. 2006). For firms competing in an industry with low profit margins, the adoption of low-cost manufacturing is critical to the financial sustainability of the firms, and also enables firms to keep prices low and reduce the instances of brand substitution.

　　However, the lean strategies adopted by fast fashion houses significantly reduces transparency in the SC. This is especially problematic nowadays, as the expectation to enforce and maintain ethical sourcing and manufacturing throughout the SC is unquestionable. In fact, firms often view ethical sourcing as a competitive advantage as it enhances global marketing, whereas lack of transparency can lead to neglect towards instances of human rights and environment exploitation (Pedersen and Andersen 2013). This can result in legal liabilities and poor brand image due to association (Plambeck 2012; Soyka 2012). Research conducted for the Nordic Council found that 83% of consumers prefer brands that exhibit transparency in their SC (Nagurney et al. 2015). When adopting lean SCM, firms should foster pre-determined contracts with their suppliers in terms of incorporating company principles and ethics for reputational protection should suppliers look to engage in unethical or environmentally irresponsible practices. Further, other methods to improve SC transparency include the implementation of standards developed by The Sustainable Apparel Coalition, such as the Higg Index which measures the social and environmental impacts induced by the SC (Westervelt 2012).

　　Much research has been completed on the effectiveness of lean SCs and it is often found that firms should optimize their own key processes prior to engaging with suppliers. For example, a case study by Kram et al. (2015) on a factory in Croatia found that when lean management tools were applied, inefficient communication systems between the firm and manufacturer was the cause for latency in the delivery of supplies rather than the suspected inefficiency of suppliers. Lack of efficient communication between entities created a *bullwhip effect*, where distorted information between entities

increased lead time, and led to poor inventory management as well as increased tension between the entities. Psychic distance is often suggested as a factor for increased costs involved in managing such a complex SC with respect to communication efficiency (Alessandro et al. 2015). The study recommended the implementation of sophisticated technology for improved communication, such as Enterprise Resource Planning systems, ICT, or blockchain, to mitigate this.

It is clear that lean SCM does not directly contribute to competitive advantage. In fact, holding reduced inventory can result in loss of sales and market share (Christopher and Towill 2000; Tang 2006) if 'ripple effects' such as the bullwhip effect become more frequent. One such example is how Boeing underestimated the volatility of demand in the aerospace industry when they adopted a lean SC strategy with a focus on reduced inventory (Naylor et al. 1999). This resulted in allowing their only competitor Airbus Industrie to better meet customer demands and gain market share. The rigid characteristics of lean SCM mean unexpected developments such as the COVID-19 pandemic may also cause drastic disruptions in the SC. While a historically predictable and stable fashion industry provided propitious conditions for applying lean SCM, modern fashion trends have moved from an annual two-season vogue within a year to accommodating to rapidly changing trends. Firms and retailers require expedited replenishment of inventory to keep up with trends, which means potential disruptions in SC or increased lead times result in the depletion of market share.

## 3.2. Agile (Flexible) Supply Chain Management

The advancement of marketing through the internet and social media influenced the growth of a *See Now-Buy Now* model in the fashion industry (Robinson and Hsieh 2016), resulting in increased volatility in demand and customer preferences. This led to the adoption and implementation of a more flexible and agile SCM to ensure firms are capable of responding to uncertainty (Christopher 2000; Prater et al. 2001). The fashion industry's evolution meant market conditions changed to pilot shorter life cycle products, high product variety, unpredictable demand, small volume with high-profit margins, as well as product differentiation as a competitive advantage (Pearson et al. 2010). This increase in volume and variety required firms to respond rapidly to consumer preferences, fostering the adoption of agile SCM (Fan et al. 2007). Much like lean SCM, agile SCM also focuses on immediate responses to urgent deliveries and order fulfillment rates (Cabral et al. 2012). However, they differ in how agile SCM is applied downstream (Mason-Jones et al. 2000) with the integration of the business and SC partners as the main focus. Using agile SCM, firms use enhanced communication systems and relationship configurations with greater visibility of information between entities throughout the SC to expedite responses to changes (Baramichai et al. 2007; Bottani 2009).

The transition of firms from cost-saving strategies to value creation improved their market positions and afforded them a significant competitive advantage. This increased the adoption of agile SC over lean SC in the industry (Bruce and Daly 2011; Caniato et al. 2011; Ellram et al. 2013). Agility also allows better coordination and management of the SC with reduced disruptions and decreased lead time (McLaren et al. 2002) due to improved information delivery. While agile SCM can improve the diversified product portfolio of fast fashion firms, in the luxury segment, brands such as Burberry and Gucci will also be able to ascertain greater control over their product quality, preserving their reputation of heritage and prestige. For these firms, R&D is important to maintaining a competitive advantage by procuring innovative designs and fabrics which can be easily and swiftly transitioned to the manufacturing stage if the firm is able to have efficient SC coordination (Fratocchi et al. 2015; Pisano and Shih 2012). Enhancing coordination and communication within SCs also facilitates greater transparency where firms are able to build better relationships with suppliers and foster ethical and environmentally-friendly practices, hence reducing reputational risks. Pearson et al. (2010) proposed the idea of Complex Adaptive Systems (CAS) to monitor supplier and intermediary networks to improve decision-making. Although this can be costly in the short-term, agile SCM may offer increased competition between suppliers in the long-term, as

firms will have a better knowledge of the supplier systems, offering potential cost savings in materials, production, and management.

A study completed by Mehrjoo and Pasek (2016) on SMEs in the manufacturing sector provided guidance on implementing effective agile SCs. Firms employing an entrepreneurial orientation with emphasis on innovation performance by building better relationships with supply partners found evidence for the sustainability of SCs (Cheng et al. 2014), improving SCM. Employing due diligence during the selection and management of suppliers (Luo et al. 2009) as well as optimized resource management, both materials, and human resources, also helped improve inventory management (Mehrjoo and Pasek 2016), delivering lean SCM attributes while maintaining agility. This means that the suppliers chosen should have the capacity to adapt quickly to market changes, thus facilitating higher profit gain during uncertainties (Naylor et al. 1999). Implementation of *Just-in-Time* can also supplement inventory management (Power et al. 2001) while the utilization of progressive technology is paramount to improving communication channels and reducing information incoherence (Pereira et al. 2015), hence controlling for risks such as the *bullwhip effect*.

The key shortcoming of agile SCM is increased inventory levels, which can increase inventory management costs and holding costs whilst raising the risk of material obsolescence (Carvalho et al. 2011). Agile SCM is based on offering the flexibility to change SC entities to suit the scenarios and situations. This also entails holding a large inventory within a SC, which can generate large losses with reduced flexibility for the firm in practice (Cabral et al. 2012). Firms who aim to apply agile SCM should decide on what level of inventory capacity would be optimal for their operations, given an acceptable increase in holding and management costs. Alternative strategies using agile SCM that many firms undertake include building alliances with other firms to share SC activities as well as leveraging synergies between different sectors of the firm, as seen with luxury fashion conglomerates.

## 4. Implications of COVID-19 for Supply Chain Risk Management

### 4.1. Supply Chain Disruption

Typical disruption risks (natural disasters, man-made catastrophes, legal disputes and/or strikes) have a strong and immediate impact on SCs. Furthermore, disruptions in one phase of the SC may propagate downstream if there are not adequate 'buffers', leading to a *ripple effect* where SC performance is increasingly compromised with each propagation of the *ripple* (Dolgui et al. 2018). Haren and Simchi-Levi (2020) highlight that SCs that are characterized by low-lead times will see disruptions propagate down the chain faster, as they typically do not have buffers.

Industries that are highly globalized are especially prone to *epidemic disruption* (Ivanov 2020). The first quarter of 2020 saw a 3% decline in global trade values with 94% of Fortune 100 companies reporting COVID-19 related SC disruptions (Teodoro and Rodriguez 2020).

Ivanov (2020) suggests epidemic outbreaks are characterized by three elements: (1) unpredictable long-term disruption, (2) the ripple effect propagating disruptions throughout the SC, and (3) concurrent disruptions in logistics networks and in supply and demand. Typical disruption risks are limited in geographical scope; however, due to today's globalized society, epidemics often start small before rapidly dispersing across vast geographical regions.

Components of the SC that were once considered independent may no longer be (Ivanov 2020). Consider a firm with a dual sourcing strategy where both suppliers are located in the same region. If an economic lockdown occurs, both suppliers may become unable to fulfill orders. The non-essential nature of the fashion industry makes it especially prone to lockdown restrictions. Furthermore, as epidemics spread in an asynchronous way and countries have varying control responses, we observe a unique dimension in *epidemic disruption*: different parts of the SC will experience varying levels of disruption at different periods, either from a *ripple* downstream, or as a direct result of local lockdowns. While China's COVID-19 lockdowns began in late-January, most European countries (excluding Italy)

did not lockdown until mid-March (BBC 2020). The varying epidemic responses and infection rates amongst countries also affect SCs. At the time of writing, SCs in Africa have faced significantly less disruption than those throughout Southeast Asia due to significantly lower infection rates (BBC 2020).

The textile and fashion industries have come to rely heavily upon developing economies for low-cost sourcing and manufacturing and are therefore particularly at risk of the aforementioned disruptions (Teodoro and Rodriguez 2020). Rising wages in China over the past decade have seen some production relocated to countries such as India, Pakistan, Vietnam, and Bangladesh (Teodoro and Rodriguez 2020). Nevertheless, China remains a critical part of the fashion SC, functioning as a vital supplier of inputs, manufacturers of more high-end goods, and as a consumer. At the global level, the fashion industry is still greatly dependent on China. For instance, 70% of woven fabrics used in Bangladesh's garment industry, and 90% of Myanmar's are sourced from China (Aung and Paul 2020). In the early months of 2020 when China implemented lockdowns, these SCs were temporarily compromised. This also highlights the geographically complex and interdependent nature of the fashion SC, which makes it prone to epidemic disruption and its consequent *ripple effect*. China's lockdown immediately impacted the nation's exports (Araz et al. 2020) and in turn, delays propagated down the SC. Conversely, as Chinese factories began to return to production and stores reopened, the ongoing lockdown in Southeast Asia, Europe, and the Americas continued to disrupt China's production capabilities. These delays are especially problematic in the fashion industry, as clothes are often sold on a seasonal basis. Comparatively, in fast fashion, where clothes are often sold on two to four-week cycles, SC disruption presents an even greater challenge (Aung and Paul 2020). The push for cost-reductions and SC efficiency has thus created fashion SCs that are highly exposed to epidemic disruption.

### 4.2. Demand Disruption

COVID-19 has also been seen to cause simultaneous disturbances in supply and demand, where changing consumer demands and consequent order cancellations have a protracted effect on the global SC (Teodoro and Rodriguez 2020; Ivanov 2020). With shops closed and consumers in lockdown, there have been significant changes in demand for fashion products. A study by Andersen et al. (2020) found that consumer spending in Denmark saw an aggregate decrease of 27% in the seven weeks following lockdowns. They proposed that the economic downturn and corresponding job losses have meant that consumer spending became increasingly directed towards essential goods. In the first two weeks of March 2020, H&M's and Zara's parent company Inditex reported a 24.1% drop in sales (Inditex 2020). As of late March, Inditex reported that the majority of its Chinese stores have reopened, however, 3785 stores across 39 international markets remain temporarily shut. Online sales similarly took a hit, although to a lesser extent, with a 4.9% contraction between 1 February to 16 March (Inditex 2020).

The low-lead time strategy deployed within the fast fashion SC is, in part, dependent upon assumptions about consumer demand (Pearson et al. 2010). As lockdowns and declining economic conditions reduce the demand for non-essential goods, many firms have been canceling orders and/or invoking 'force majeure' clauses (Teodoro and Rodriguez 2020).

Again, we see a scenario where epidemics can compromise both upstream and downstream components of the SC.

### 5. Frontier Issues: Balancing Risk, Cost, and Agility Post-COVID-19

COVID-19 has revealed a new set of SCM risks for multinationals to consider. The lean SC, which relies on *Just-in-Time* and *zero inventory* management strategies, is overexposed to epidemic disruption. However, building agility is an expensive exercise and it would be impractical for firms to completely overhaul their SCs to manage 'black-swan' events such as the COVID-19 pandemic. This section thus explores a variety of strategies of varying costs that firms can employ to mitigate SC and demand disruption brought on by COVID-19 in both the short- and long-term.

### 5.1. Managing Supply Chain Disruption

Given the potential for outbreaks to disrupt input-sourcing, managers should consider adjusting the sourcing mix to better diversify risk. Dual-sourcing strategies, where both suppliers are located within close geographical proximity of one another, are exposed to greater lockdown disruptions. Similarly, firms with geographically diverse networks of suppliers are still exposed to SC disruption if a product relies on inputs from multiple suppliers as a single disruption can have a consequent ripple effect. Management must determine which of its products are particularly exposed to single-source dependencies or single location dependencies and look to build appropriate risk management strategies. In the short term, this could include reallocating inventory across regions or reducing dependence on products at risk of disruption. For example, ASOS, an online-only fast fashion company, worked with suppliers to shift production to suit the new demand for loungewear and activewear (ASOS 2020). In the medium-term firms can look to build 'buffers' to mitigate the *ripple effect* when a single supplier is compromised. This can be done in two main ways: (1) firms can create an *inventory buffer* or 'safety-stock' of essential components and products and, (2) firms can create a *time buffer* by delaying the production of goods where demand is unpredictable.

Firms with critical suppliers in financial distress are incentivized to support these suppliers because they may be difficult and/or expensive to replace. Achille and Zipser (2020) highlight that if Italy's *façonniers* (contract manufacturers, typically small-scale and family-operated) become insolvent, years of craftsmanship experience will be lost and the luxury fashion SC will be compromised. Similarly, many textile and garment producers in developing countries are struggling to cope with reduced capacity, reduced demand, and order cancelations precipitated by COVID-19 (Teodoro and Rodriguez 2020). Based on the joint statement from Sustainable Textile of the Asian Region (SIAR), "Support business partners on supply chain as much as possible, and aim at long-term strategy of business continuity, supply chain unity and social sustainability", we suggest firms should (1) quantify these supports for the recovery of suppliers and (2) identify where to implement these supports for support benefit maximization (STAR 2020).

It is also important that fashion MNCs work with suppliers to ensure production facilities and factories adhere to the new social distancing and sanitation requirements. This is from both a corporate social responsibility (CSR) perspective and a financial one. While social distancing will likely see decreased factory outputs and increased costs, the risk of total factory shutdown will be minimized. Once an employee has tested positive for COVID-19, firms should act quickly and decide whether their employees should be working from home or continuing to go into a workplace (Cohn 2020). The Centre for Disease Control recommends employees who have been in 'close contact' (within six feet for a prolonged period) with the infected should isolate for at least two weeks or until a test can be completed (Cohn 2020). For example, firms should identify your employees as close contacts and advise them to work from workplace to self-isolate. For a factory where social distancing is followed, it may only have to isolate a small segment of its workforce. Comparatively, a factory without social distancing would likely have to isolate its entire workforce resulting in a complete factory shutdown. Firms should additionally develop protocols that enable them to act quickly following infection to reduce the risk of further spread.

### 5.2. Managing Demand Disruption

As mentioned, the demand for fashion goods has seen a dramatic drop due to the impacts of COVID-19. Luxury sales are up to 70% lower for this year's spring fashion season in comparison to last year (Achille and Zipser 2020). While demand is poised to return as lockdowns restrictions are eased, it is unknown to what extent it will return. With people increasingly working from home and going out less frequently, it can be assumed that the nature of the demand for certain fashion products will change. Furthermore, the potential for, and scale of, 'second-waves' of COVID-19 infections is still largely unknown. Firms should, therefore, consider a number of strategies to manage future demand disruption.

In the early months of 2020, The Kering Group reallocated inventory from China to less affected parts of the world. The firm's SCM strategy involves replenishing product supplies as they are sold, as opposed to shipping all inventory to regions at the beginning of each fashion season (Leonard 2020). This has granted the group a high degree of flexibility in responding to demand changes, as products did not become 'trapped' in China during the initial outbreak. By June 2020, with China's economy reopening, the firm can look to replenish its Chinese stock levels. As such, fashion companies should consider restocking throughout the fashion season in response to actual demand as opposed to at the start of the season based on predicted demand. This SCM strategy yields greater agility in responding to potential demand disruptions.

Comparatively, firms with a strong online presence should be able to sustain sales after brick-and-mortar storefronts are closed. ASOS, which is an online-only retailer, reported canceling less than 1% of its Spring/Summer 2020 intake (ASOS 2020). Online product offerings are also easier to update than physical ones, therefore firms with online stores can rapidly adjust their product offering to suit new demand conditions. In the long-term, all fashion firms should look to enhance their online presence. Achille and Zipser (2020) contend that luxury companies should place particular focus on the development of 'the digital experience' given their target market is accustomed to a high quality of service in stores and expects the same online. This can be done internally by investing in proprietary online storefronts, and/or through alliances with reputable online retailers. Recent developments in augmented-reality technology can also be adapted to create digital showrooms and fitting rooms. GAP unveiled its *'DressingRoom'* app at the 2018 CES convention, which allows customers to virtually try on clothing (Sheldon 2019). Similarly, Warby Parker, a seller of eyeglasses and sunglasses, has integrated a virtual try-on feature into its iPhone app (Warby Parker n.d.). Firms should thus look for innovative ways to develop their online presence in the post COVID-19 business environment to mitigate demand disruptions by optimizing the consumer experience.

Research on COVID-19 and its implications for global SCM is still relatively scarce. Further, the long-term impact of COVID-19 on SCs and on consumer demand still remains to be seen. As firms release quarterly earnings and reports, and as more information about mitigating COVID-19 transmission comes to light, both academics and business leaders should look to reassess SC strategies and SC risk management. Research could be carried out to examine the impact of epidemic disruption with numerical data, providing more compelling insights into the extent of disruption firms may have experienced. This paper mostly neglected the CSR component of global SCM. As seen in Section 5, many small and/or family-operated suppliers are facing insolvency. From a CSR perspective, multinationals should look to support these suppliers and also to ensure that working conditions overseas are COVID safe. Additionally, an emerging issue is whether COVID-19 pandemic will see a 'second wave' of transmission globally, and how this might further affect SCs and consumer buying patterns. Finally, the impacts of multilateral tensions catalyzed by disputes over COVID-19 origins on SC networks should be addressed to examine the extent of influence politics has on SCM and corporate performance. Overall, as new information about COVID-19 emerges, academics and business leaders may need to reconsider SCM strategies.

## 6. Conclusions

Following the industrialization of today's key economies, research into SCM structure primarily focused on addressing increasingly demanding consumers by moving production to nations with low-cost inputs, which has been primarily conducted through lean SCM. More recent research identifies a need to consider other forms of management in order to combat key challenges that pose a threat to corporate activity. Our review of existing literature illustrated the need for further research into the importance of flexibility in SCM, a factor often overlooked in favor of focusing on supplier-buyer relationships. This is especially important in light of the notion that COVID-19 has triggered widespread SC disruption in various nations across different points in time. Previous research has offered key arguments both in support of and against complex SCs. While the flexibility and diversification benefits

offered by complex structures do assist in hedging risk, it is at the cost of diminishing economies-of-scale if firms over-diversify, as well as risking greater susceptibility to the resonance of disruption effects along with the entire SC. Through our review, it became clear that the optimal SCM structure largely depends on firm-specific characteristics.

Research into the merits and faults of lean compared to agile SCs also sheds light on this notion. Lean SCM has been widely adopted by fast fashion firms with its emphasis on cost and waste minimization. However, this significantly reduces transparency in SCs, which has historically resulted in widespread and highly publicized backlash for many firms. Further, the rigidness inherent in lean SCM entails that negative surprises can cause drastic disruptions to the SC. Agile SCM addresses this inflexibility by taking inherent uncertainty into consideration.

Through superior communication mechanisms and greater emphasis on the efficient transfer of information between suppliers and buyers, partnerships are enhanced to accelerate responses to changes in the operating environment. Thus, the focus is shifted from cost-saving in lean SCM to value-creation in agile SCM. The latter, however, is limited by increased inventory needs which entails greater management and holding costs.

Existing research into SCM has yet to address how COVID-19 has impacted SCs and how this has influenced SCM theory. The COVID-19 pandemic has highlighted risks in both concentrated and global SCs as concentrated sourcing is greatly susceptible to epidemic disruption risk while global SCs has seen different parts of the SC shut down at different times as countries vary in when the virus spreads and their capacity to contain it. This has been especially problematic in the fashion industry due to the seasonality of output and the interdependent nature of the SC. Furthermore, the fashion industry has seen demand disruption due to widespread lockdowns, as well as an economic downturn which yields lower consumption. The risks in SCM that have arisen due to the COVID-19 outbreak highlights the need for further research and guidance as to how to build agility to mitigate epidemic-induced risks.

To incorporate greater agility into SCs, corporations can consider implementing risk management strategies that take sourcing dependencies into consideration and introduce 'buffers' to reduce the impacts of disruptions. Additionally, firms should look to assist their key suppliers who are facing insolvency and ensure social distancing restrictions are adhered to in order to protect long-term interests. Stocking strategies based on actual demand rather than predicted may yield greater flexibility for some firms, and investments into online customer experience need to be made to control for demand disruptions and maximize sales.

In order to expand our knowledge of SC risk management, further research is needed as more information on the effects of the pandemic on international business is published. We identified four key research areas: (1) quantitative modeling of the SC disruption caused by the COVID-19 pandemic, (2) initiatives firms undertake to assist financially distressed suppliers, (3) the impacts of a potential second-wave of infections on consumption patterns and SCM, and (4) the influence of political disputes on international business and SCM. Overall, this paper illustrates some key issues that have been drawn out by the COVID-19 pandemic whereby scholars and business leaders will need to rethink fashion SCM moving forwards.

**Author Contributions:** Conceptualization, M.M., C.N., C.T., B.X., C.C., and P.Q.; resources, M.M., C.N., C.T., B.X., C.C., and P.Q.; writing—original draft preparation, M.M., C.N., C.T., B.X., and C.C.; writing—review and editing, M.M., C.N., C.T., B.X., C.C., and P.Q. All authors have read and agreed to the published version of the manuscript.

**Funding:** This research received no external funding.

**Conflicts of Interest:** The authors declare no conflict of interest.

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
