# Peer review of "Risk Management: Rethinking Fashion Supply Chain Management for Multinational Corporations in Light of the COVID-19 Outbreak"

_jrfm, doi:10.3390/jrfm13080173_

Round 1

Reviewer 1 Report

The authors reviewed SC risk management issues in light of the COVID-19 outbreak to provide future research direction and guidance on building agility after COVID-19.  The paper is well written.

They assert that COVID-19 pandemic posed to catalyze a movement towards more agile SCs. They discuss the merits and limitations of lean versus flexible supply chains (SC) and contend that fashion supply chains should consider the long-term structural changes that should last after COVID-19.

I had difficulty finding the contribution of this article. I think authors reviewed well what happened after COVID 19 and give some recommendation based on what they read. Since there are not much data it is understandable. So, I decided that article can be published if the authors address the following comments.

Authors contend that “In the medium-term firms can look to build ‘buffers’ to mitigate the ripple effect when a single supplier is compromised. This can be done in two main ways: (1) firms can create an inventory buffer or ‘safety-stock’ of essential components and products and, (2) firms can create a time buffer by delaying the production of goods where demand is unpredictable.” This recommendation sounds reasonable. It might be helpful some practitioners.

Authors assume that lean supply chain management lack of supply chain transparency. I do not see the relationship between transparency and whether SC is lean or flexible. Transparency may not exist in flexible supply chains either.  

I am quite surprised that authors talk about the bullwhip effect in lean SCM, though they site (Snyder & Shen, 2019). I agree with Sarkar and Kumar (2015) that the bullwhip effect can be mitigated by sharing real-time information with today’s technology. I am astonished why Snyder & Shen missed this study and why the authors did not critique study of Snyder & Shen. Do I miss something? If the firms do not have ERP it is quite understandable that there will be bullwhip affect. So, it is not because they are lean of flexible it is because they do not have integrated information System (ERP). Authors recommend “the implementation of sophisticated technology for improved communication, such as Enterprise Resource Planning systems, to mitigate bullwhip effect.” But it is already recommended by Sarkar and Kumar.

They recommend “Firms should, therefore, avoid canceling order and invoking force-majeure clauses where possible. They should look to support the recovery of critical suppliers, as the long-term disruption of these suppliers becoming obsolete will significantly outweigh the temporary epidemic disruption.” Yet there is no calculation. Key word is “where possible.” How a firm would know if it is possible or not. Mostly opinion not much data driven recommendation.

Authors wrote “While social distancing will likely see decreased factory outputs and increased costs, the risk of total factory shutdown will be minimized.” These are “center for disease control” recommendations. I am not sure if any of these will be helpful neither for academia nor practitioners. Authors seem to support what center for disease control recommends.

Authors wrote “Through our review, it became clear that the optimal SCM structure largely depends on firm-specific characteristics.” I agree with them that optimal SCM structure largely depends on firm specific characteristics that is why for some SCs lean for some other flexible structure is recommended. That is well known fact.

Authors recommend “Additionally, firms should look to assist their key suppliers who are facing insolvency and ensure social distancing restrictions are adhered to in order to protect long-term interests. Stocking strategies based on actual demand rather than predicted may yield greater flexibility for some firms, and investments into online customer experience need to be made to control for demand disruptions and maximize sales.” These are good recommendations yet not much data driven. I understand there is not much data yet.

They recommend “(1) quantitative modelling of the SC disruption caused by the COVID-19 pandemic, (2) initiatives firms undertake to assist financially distressed suppliers, (3) the impacts of a potential second-wave of infections on consumption patterns and SCM and (4) the influence of political disputes on international business and SCM.”

That is what I was expecting from this paper some data driven recommendations. The most interesting recommendation, in my judgment, is “the influence of political disputes on international business and SCM.” Yet I am not sure how it could be measured how it could be isolated from other factors.

Author Response

I really appreciate your time and so constructive suggestions.

Ref: Manuscript ID: jrfm-861287     

Title: Risk Management: Rethinking Fashion Supply Chain Management for Multinational Corporations in Light of the COVID-19 Outbreak

Journal: Journal of Risk and Financial Management

This document contains our responses to the comments of the review team on Manuscript jrfm-861287. We are really grateful to the reviewers for their constructive suggestions, which help us significantly improve our paper. Throughout this document, we italicize the comments of the review team.

Response to Reviewers

We thank you for your constructive feedback and suggestions. After considering all the useful and constructive comments of the review team, we revise the paper accordingly.

Response to Reviewer:

Major comments

The authors reviewed SC risk management issues in light of the COVID-19 outbreak to provide future research direction and guidance on building agility after COVID-19.  The paper is well written.

They assert that COVID-19 pandemic posed to catalyze a movement towards more agile SCs. They discuss the merits and limitations of lean versus flexible supply chains (SC) and contend that fashion supply chains should consider the long-term structural changes that should last after COVID-19.

I had difficulty finding the contribution of this article. I think authors reviewed well what happened after COVID 19 and give some recommendation based on what they read. Since there are not much data it is understandable. So, I decided that article can be published if the authors address the following comments.

  1. Authors assume that lean supply chain management lack of supply chain transparency. I do not see the relationship between transparency and whether SC is lean or flexible. Transparency may not exist in flexible supply chains either.

Response

Thanks for your comments and suggestions. Following your suggestions, we explain more about the transparency (Refer to p. 1, line 22-23 of the abstract ) as below:

“Although this problem might exist in the agile supply chain, agile supply chains combat this by focusing on enhancing communication and buyer-supplier relationships to improve information exchange.”

  1. I am quite surprised that authors talk about the bullwhip effect in lean SCM, though they site (Snyder & Shen, 2019). I agree with Sarkar and Kumar (2015) that the bullwhip effect can be mitigated by sharing real-time information with today’s technology. I am astonished why Snyder & Shen missed this study and why the authors did not critique study of Snyder & Shen. Do I miss something? If the firms do not have ERP it is quite understandable that there will be bullwhip affect. So, it is not because they are lean of flexible it is because they do not have integrated information System (ERP). Authors recommend “the implementation of sophisticated technology for improved communication, such as Enterprise Resource Planning systems, to mitigate bullwhip effect.” But it is already recommended by Sarkar and Kumar.

Response

Thanks for your comments and suggestions. As for bullwhip effect, We delete the “lean”, as bullwhip effect is a common issue happening in SCM, not just in a lean supply chain but in a flexible chain. Additionally, through review literature in the last 5 years, particularly papers after the COVID-19. We also supplement discussion on the ripple effect. (Refer to p. 4, 40-43 of the 2.3. Supply Chain Disruption Risk and Mitigation Strategies)

“Issues experienced by SCM are the bullwhip effect and ripple effect….. while the latter occurs when a disruption triggers a chain effect downstream and upstream which influences SC performance (Bevilacqua et al., 2019).”

“To reduce the ripple effect, Ivanov et al. (2019) investigate the role of digital technologies and Industry 4.0 in rising demand responsiveness and capability flexibility. With the support of big data analytics (BDA) and tracking and tracing system (T&T) technologies, Industry 4.0 increases the ability to reconfigure resources at the recovery stage (Ivanov et al., 2019).”

References

Bevilacqua, M., Ciarapica, F. E., Marcucci, G., & Mazzuto, G. (2019). Fuzzy cognitive maps approach for analysing the domino effect of factors affecting supply chain resilience: A fashion industry case study. International Journal of Production Research, 1-29.

Ivanov, D., Dolgui, A., & Sokolov, B. (2019). The impact of digital technology and Industry 4.0 on the ripple effect and supply chain risk analytics. International Journal of Production Research, 57(3), 829-846.

As for sophisticated technology recommendations, we supplement examples of ICT or blockchain (Refer to p.6, line 29 of the 3.1. Lean Supply Chain Management).

“The study recommended the implementation of sophisticated technology for improved communication, such as Enterprise Resource Planning systems, ICT, or blockchain, to mitigate this.”

  1. They recommend “Firms should, therefore, avoid canceling order and invoking force-majeure clauses where possible. They should look to support the recovery of critical suppliers, as the long-term disruption of these suppliers becoming obsolete will significantly outweigh the temporary epidemic disruption.” Yet there is no calculation. Key word is “where possible.” How a firm would know if it is possible or not. Mostly opinion not much data driven recommendation.

Response

Thanks for your comments. We refer to the joint statement from Sustainable Textile of the Asian Region, “Support business partners on supply chain as much as possible, and aim at long-term strategy of business continuity, supply chain unity and social sustainability”, and then modify our recommendation (Refer to p.10, line 16-20 of the 5.1.Managing Supply Chain Disruption) as below:

“Based on the joint statement from Sustainable Textile of the Asian Region (SIAR), “Support business partners on supply chain as much as possible, and aim at long-term strategy of business continuity, supply chain unity and social sustainability”, we suggest firms should (1) quantify these supports for the recovery of suppliers and (2) identify where to implement these supports for support benefit maximization.”

References

STAR. (2020). Joint Statement on Responsible Purchasing Practices amid the COVID-19 Crisis[online]. Available from: http://www.asiatex.org/ennewss/393.html [accessed 18 July 2020]

  1. Authors wrote “While social distancing will likely see decreased factory outputs and increased costs, the risk of total factory shutdown will be minimized.” These are “center for disease control” recommendations. I am not sure if any of these will be helpful neither for academia nor practitioners. Authors seem to support what center for disease control recommends.
  2. Authors recommend “Additionally, firms should look to assist their key suppliers who are facing insolvency and ensure social distancing restrictions are adhered to in order to protect long-term interests. Stocking strategies based on actual demand rather than predicted may yield greater flexibility for some firms, and investments into online customer experience need to be made to control for demand disruptions and maximize sales.” These are good recommendations yet not much data-driven. I understand there is not much data yet.

Response

Thanks for your comments. This recommendation includes two waves: centralized isolation or separated isolation, depending on specific situations. Therefore, we have added some explanations in the revised paper as below: (Refer to p. 10, line 20-21 and line 24-25 of the 5.1.Managing Supply Chain Disruption)

“Once find an employee tested positive for COVID-19, firms should act quickly and decide whether your employees are working from home or continuing to go into a workplace (Cohn, 2020).”

“For example, firms should identify your employees as close contacts and advice them to work from workplace to self-isolate.”

  1. They recommend “(1) quantitative modelling of the SC disruption caused by the COVID-19 pandemic, (2) initiatives firms undertake to assist financially distressed suppliers, (3) the impacts of a potential second-wave of infections on consumption patterns and SCM and (4) the influence of political disputes on international business and SCM.”

That is what I was expecting from this paper some data-driven recommendations. The most interesting recommendation, in my judgment, is “the influence of political disputes on international business and SCM.” Yet I am not sure how it could be measured how it could be isolated from other factors.

Response

Thanks for your comments. We really regret for not much data-driven and try our best to use the literature to drive our recommendations. Through reviewing papers about supply chain risk mitigation after COVID-19, there are three articles from Ivanov. Paper “Predicting the impacts of epidemic outbreaks on global supply chains: A simulation-based analysis on the coronavirus outbreak (COVID-19/SARS-CoV-2) case” uses simulation experiments to identify the successful and wrong elements of risk mitigation/preparedness and recovery policies, which can support to our recommendations (1) (3) to some extent.

As for (2), we based on the joint statement from Sustainable Textile of the Asian Region (SIAR): “Based on the joint statement from Sustainable Textile of the Asian Region (SIAR), “Support business partners on supply chain as much as possible, and aim at long-term strategy of business continuity, supply chain unity and social sustainability”, we suggest firms should (1) quantify these supports for the recovery of suppliers and (2) identify where to implement these supports for support benefit maximization.”

Concerning “the influence of political disputes on international business and SCM”, it might be a topic for future opportunities. Whether can we learn “government support for innovation” who uses subsidies to measure government support and financial factors, such as net profit, increase rate of business revenue, etc., to test innovation performance? How about using political actions as a moderator to measure the relations between inventory and demand? However, this is just an idea which should be measured its feasibility. But I strongly appreciate your suggestions that inspire my idea.

Reviewer 2 Report

This paper reviewed previous research in terms of supply chain risk mitigation strategies. It is very proper to suggest or review previous well-known strategies after COVID-19 outbreak. 

However, as a reviewer, I could not find any valuable insights from the script. 

First of all, I kindly suggest authors develop a framework to review previous works, find limitations, and suggest some insights. 

Also, please make clear what authors want to convey to readers. 

After reading previous research, please revise the structure of the whole script. 

Author Response

I really appreciate your time and so constructive suggestions. 

Ref: Manuscript ID: jrfm-861287     

Title: Risk Management: Rethinking Fashion Supply Chain Management for Multinational Corporations in Light of the COVID-19 Outbreak

Journal: Journal of Risk and Financial Management

This document contains our responses to the comments of the review team on Manuscript jrfm-861287. We are really grateful to the reviewers for their constructive suggestions, which help us significantly improve our paper. Throughout this document, we italicize the comments of the review team.

Response to Reviewers

We thank you for your constructive feedback and suggestions. After considering all the useful and constructive comments of the review team, we revise the paper accordingly.

Response to Reviewer:

Major comments

This paper reviewed previous research in terms of supply chain risk mitigation strategies. It is very proper to suggest or review previous well-known strategies after COVID-19 outbreak.

However, as a reviewer, I could not find any valuable insights from the script.

First of all, I kindly suggest authors develop a framework to review previous works, find limitations, and suggest some insights.

Also, please make clear what authors want to convey to readers.

After reading previous research, please revise the structure of the whole script.

Response

Thanks for your encouragement and constructive comments. We revise the introduction by (1) focusing on risks of fashion industry and the effects pandemic, (2)supplementing references in last 6 years, and (3) sorting out structures of the article(Refer to p.2, line 11-18 of the Introduction).

Then, we merge the previous section 6 into section 5.

“Since the adoption of global SCs, fashion industry can be featured in terms of volatility, velocity, variety, complexity and dynamism (ÄŒiarnienÄ— et al., 2014; Mustafid et al., 2018), and SC in the fashion sector is full of uncertainty and unpredictability (Giannakis and Louis, 2016; Mustafid et al., 2018). Fashion multinational corporations (MNC) have been subject to key risks that have often been realised, such as late deliveries, long lead time between returns and resending to customers, stock-out or over-stock, and deliveries in single solutions (Martino et al., 2015). Moreover, the internationalisation of SCs has exposed fashion MNCs to country-specific risks, which leads local high-risk events to snowball in magnitude (Ivanov, 2018; Bevilacqua et al., 2019). Due to the COVID-19 outbreak, in the midst of the pandemic with countries varying in their methods and capacity to contain the virus, social distancing and lock-down requirements limiting production in key supplier markets reveal the shortcomings of existing supply chain management (SCM) models.”

“Section 2 holistically review and expand on SC risk management issues in light of the COVID-19 outbreak and an overview of theory underpinning modern SCM. In section 3, we discuss the merits and faults of lean and agile SCM styles before analysing the risks highlighted and drawn out in the COVID-19 period. Based on background and theoretical review, section 4 is the implications of COVID-19 for SC risk management. We provide front issues in section 5 while the conclusion is shown in section 6.”

References

Čiarnienė, R., Vienažindienė, M. (2014). Agility and responsiveness managing fashion supply chain. Procedia-Social and Behavioral Sciences,150(Supplement C), 1012-1019.

Mustafid, Karimariza, S. A., Jie, F. (2018). Supply chain agility information systems with key factors for fashion industry competitiveness. International Journal of Agile Systems and Management, 11(1), 1-22.

Giannakis, M., & Louis, M. (2016). A multi-agent based system with big data processing for enhanced supply chain agility. Journal of Enterprise Information Management.

Martino, G., Fera, M., Iannone, R., Sarno, D., & Miranda, S., Senigallia, Italy. (2015). Risk identification map for a fashion retail supply chain. Proceedings of Summer School “Francesco Turco”, Senigallia, Italy, 208-216.

Ivanov, D. (2018). Revealing interfaces of supply chain resilience and sustainability: a simulation study. International Journal of Production Research, 56(10), 3507-3523.

Bevilacqua, M., Ciarapica, F. E., Marcucci, G., & Mazzuto, G. (2019). Fuzzy cognitive maps approach for analysing the domino effect of factors affecting supply chain resilience: A fashion industry case study. International Journal of Production Research, 1-29.

Reviewer 3 Report

Thank you for sending me the paper "Risk Management: Rethinking Fashion Supply Chain Management for Multinational Corporations in Light of the COVID-19 Outbreak".

Though the methodology is well developed and presented, I have to put the attention on the literature cited in the paper.

Literature review is at the basis of this study, but the literature cited in the paper is sometimes not up to date.

1 Introduction: the introduction should lay the basis of all the study, and there are few papers that are definitevely not up to date (2002, 2002, 2009, 2015)

2.2 Supply Chains in the Fashion Industry: here too, the literature is old.

2.3: Supply Chain Disruption Risk and Mitigation Strategies: i suggest the authors to focus only on strategies laid out during the last 5 years, so all the literature before 2015 should be reconsidered.

In general the paper must be improved by citing all the relevant and recent research. I can suggest a few:

Bevilacqua, M., F. E. Ciarapica, G. Marcucci, and G. Mazzuto. 2018a. “Conceptual Model for Analysing Domino Effect among Concepts
Affecting Supply Chain Resilience.” Supply Chain Forum: An International Journal 19 (4): 282–299.

Bevilacqua, M., Ciarapica, F. E., Marcucci, G., & Mazzuto, G. (2019). Fuzzy cognitive maps approach for analysing the domino effect of factors affecting supply chain resilience: A fashion industry case study. International Journal of Production Research, 1-29.

Li, W.-Y., P.-S. Chow, T.-M. Choi, and H.-L. Chan. 2016. “Supplier Integration, Green Sustainability Programs, and Financial
Performance of Fashion Enterprises Under Global Financial Crisis.” Journal of Cleaner Production 135: 57–70.

Martino, G., M. Fera, R. Iannone, D. Sarno, and S. Miranda. 2015. “Risk Identification Map for a Fashion Retail Supply Chain.” In:
Proceedings of the Summer School “Francesco Turco” Senigallia Italy 208–216.

Mehrjoo, M., and Z. J. Pasek. 2016. “Risk Assessment for the Supply Chain of Fast Fashion Apparel Industry: A System Dynamics
Framework.” International Journal of Production Research 54: 28–48.

Mustafid, Karimariza, S. A., and F. Jie. 2018. “Supply Chain Agility Information Systems with Key Factors for Fashion Industry
Competitiveness.” International Journal of Agile Systems and Management 11 (1): 1–22.

Author Response

I really appreciate your time and so constructive suggestions. 

Ref: Manuscript ID: jrfm-861287     

Title: Risk Management: Rethinking Fashion Supply Chain Management for Multinational Corporations in Light of the COVID-19 Outbreak

Journal: Journal of Risk and Financial Management

This document contains our responses to the comments of the review team on Manuscript jrfm-861287. We are really grateful to the reviewers for their constructive suggestions, which help us significantly improve our paper. Throughout this document, we italicize the comments of the review team.

Response to Reviewers

We thank you for your constructive feedback and suggestions. After considering all the useful and constructive comments of the review team, we revise the paper accordingly.

Response to Reviewer:

Major comments

Thank you for sending me the paper "Risk Management: Rethinking Fashion Supply Chain Management for Multinational Corporations in Light of the COVID-19 Outbreak".

Though the methodology is well developed and presented, I have to put the attention on the literature cited in the paper.

Literature review is at the basis of this study, but the literature cited in the paper is sometimes not up to date.

  1. Introduction: the introduction should lay the basis of all the study, and there are few papers that are definitevely not up to date (2002, 2002, 2009, 2015).

Response

Thanks for your encouragement and constructive comments. We revise the introduction by (1) focusing on risks of fashion industry and the effects pandemic, (2)supplementing references in last 6 years, and (3) sorting out structures of the article. (Refer to p.2, line 11-18 of the Introduction)

“Since the adoption of global SCs, fashion industry can be featured in terms of volatility, velocity, variety, complexity and dynamism (ÄŒiarnienÄ— et al., 2014; Mustafid et al., 2018), and SC in the fashion sector is full of uncertainty and unpredictability (Giannakis and Louis, 2016; Mustafid et al., 2018). Fashion multinational corporations (MNC) have been subject to key risks that have often been realised, such as late deliveries, long lead time between returns and resending to customers, stock-out or over-stock, and deliveries in single solutions (Martino et al., 2015). Moreover, the internationalisation of SCs has exposed fashion MNCs to country-specific risks, which leads local high-risk events to snowball in magnitude (Ivanov, 2018; Bevilacqua et al., 2019). Due to the COVID-19 outbreak, in the midst of the pandemic with countries varying in their methods and capacity to contain the virus, social distancing and lock-down requirements limiting production in key supplier markets reveal the shortcomings of existing supply chain management (SCM) models.”

“Section 2 holistically review and expand on SC risk management issues in light of the COVID-19 outbreak and an overview of theory underpinning modern SCM. In section 3, we discuss the merits and faults of lean and agile SCM styles before analysing the risks highlighted and drawn out in the COVID-19 period. Based on background and theoretical review, section 4 is the implications of COVID-19 for SC risk management. We provide front issues in section 5 while the conclusion is shown in section 6.”

References

Čiarnienė, R., Vienažindienė, M. (2014). Agility and responsiveness managing fashion supply chain. Procedia-Social and Behavioral Sciences,150(Supplement C), 1012-1019.

Mustafid, Karimariza, S. A., Jie, F. (2018). Supply chain agility information systems with key factors for fashion industry competitiveness. International Journal of Agile Systems and Management, 11(1), 1-22.

Giannakis, M., & Louis, M. (2016). A multi-agent based system with big data processing for enhanced supply chain agility. Journal of Enterprise Information Management.

Martino, G., Fera, M., Iannone, R., Sarno, D., & Miranda, S., Senigallia, Italy. (2015). Risk identification map for a fashion retail supply chain. Proceedings of Summer School “Francesco Turco”, Senigallia, Italy, 208-216.

Ivanov, D. (2018). Revealing interfaces of supply chain resilience and sustainability: a simulation study. International Journal of Production Research, 56(10), 3507-3523.

Bevilacqua, M., Ciarapica, F. E., Marcucci, G., & Mazzuto, G. (2019). Fuzzy cognitive maps approach for analysing the domino effect of factors affecting supply chain resilience: A fashion industry case study. International Journal of Production Research, 1-29.

  1. 2.2 Supply Chains in the Fashion Industry: here too, the literature is old.

Response

Thanks for your comments and suggestions. We supplement four papers, four of which are recommended by our reviewer.

Paper 1 from Mehrjoo and Pasek (2016) focuses on the effect of lead time on SC performance, which supports our topic sentence. (Refer to p. 3, line 30-32 of the 2.2. Supply Chains in the Fashion Industry)

“The effect of lead time on SC performance (inventory, cost, backlog and risk) is the key to success for fast fashion sectors (Mehrjoo and Pasek, 2016).”

Paper 2 and paper 3 from Martino et al. (2015) and Li et al. (2016) focus on supplier integration can significantly promote financial performance, which develops our narrative “developing relationships with the outsourced fashion SC”. (Refer to p. 3, line 33-36 of the 2.2. Supply Chains in the Fashion Industry).

“Martino et al. (2015) and Li et al. (2016) argued that supplier integration can significantly promote financial performance and help to mitigate the negative effect of financial tsunami on financial performance of the fashion enterprises.”

Paper 4 and paper 5 from Gligor et al., (2015) and Mustafid et al. (2018) discuss the challenges (dynamic market and market uncertainty) in fashion SC, which are used to draw forth the importance of an agile SC. (Refer to p. 3, line40-43 of the 2.2. Supply Chains in the Fashion Industry).

“Challenges in fashion SCs areis the high demand volatility of customers (Choi 2006; 2007), dynamic market and market uncertainty (Gligor et al., 2015, Mustafid et al., 2018), which can create problems of either overstocking or understocking (Choi 2006; 2007).”

References

Gligor, D. M., Esmark, C. L., & Holcomb, M. C. (2015). Performance outcomes of supply chain agility: when should you be agile? Journal of Operations Management, 33, 71-82.

Li, W.-Y., Chow, P.-S., Choi, T.-M., & Chan, H.-L. (2016). Supplier integration, green sustainability programs, and financial performance of fashion enterprises under global financial crisis. Journal of Cleaner Production, 135, 57-70.

Martino, G., Fera, M., Iannone, R., Sarno, D., & Miranda, S., Senigallia, Italy. (2015). Risk identification map for a fashion retail supply chain. Proceedings of Summer School “Francesco Turco”, Senigallia, Italy, 208-216.

Mehrjoo, M., & Pasek, Z. (2016). Risk assessment for the supply chain of fast fashion apparel industry: a system dynamics framework. International Journal of Production Research, 54(1), 28-48.

Mustafid, Karimariza, S. A., Jie, F. (2018). Supply chain agility information systems with key factors for fashion industry competitiveness. International Journal of Agile Systems and Management, 11(1), 1-22.

  1. 2.3: Supply Chain Disruption Risk and Mitigation Strategies: i suggest the authors to focus only on strategies laid out during the last 5 years, so all the literature before 2015 should be reconsidered.

Response

Thanks for your comments and suggestions. We review literature in the last 5 years, particularly papers after the COVID-19. Papers pay attention to SC risk mitigation strategies and resilience. Our work investigates the factors on SC performance, flexibility to hedge risks, appropriate SC structure, discussion on the ripple effect. (Refer to p. 4, line 7-10, 12-17, 21-23, 26-30, 31, 33-35, 40-43 of the 2.3.  Supply Chain Disruption Risk and Mitigation Strategies)

“Ivanov and Review (2020) uses simulation-based analysis to research the impacts of COVID-19 on global SCs, highlighting its impact on factors essential to SC performance including the timing of closing and opening of the facilities at various degrees, lead time, speed of epidemic propagation, and the upstream and downstream disruption durations.”

“By researching the domino effect of factors affecting supply chain resilience (SCR) in fashion industry, Bevilacqua et al. (2019) suggest that due to manufacturers highlighting flexibility in order fulfillment, a flexible production structure is vital to effectively address unpredictable turnarounds of the market in a timely manner. Likewise, in the research note about COVID-19 and SCR, Ivanov and Das (2020) indicate that the focus of SC resilience management should alter towards situational responses to real-time changes.”

“In regards to the appropriate SC structure, which implies SC structure adaptation and flexibility, severe disruptions can change the SC structure and are involved with SC structural dynamics (Ivanov and Dolgui, 2019, Ivanov and Dolgui, 2020).”

“Motivated by COVID-19 outbreak, Ivanov and Dolgui (2020) propose the integrity of the intertwined supply network (ISN) and viability. An ISN is an entirety of interconnected SCs that secure the supply of society and markets with goods and services, while SC viability management would rather be altered towards the situational reactions (Ivanov and Das, 2020, Ivanov and Dolgui, 2020).”

“Issues experienced by SCM are the bullwhip effect and ripple effect….. while the latter occurs when a disruption triggers a chain effect downstream and upstream which influences SC performance (Bevilacqua et al., 2019).”

“To reduce the ripple effect, Ivanov et al. (2019) investigate the role of digital technologies and Industry 4.0 in rising demand responsiveness and capability flexibility. With the support of big data analytics (BDA) and tracking and tracing system (T&T) technologies, Industry 4.0 increases the ability to reconfigure resources at the recovery stage (Ivanov et al., 2019).”

References

Ivanov, D. (2020). Predicting the impacts of epidemic outbreaks on global supply chains: A simulation-based analysis on the coronavirus outbreak (COVID-19/SARS-CoV-2) case. Transporation Research Part E, 136, 1-14.

Bevilacqua, M., Ciarapica, F. E., Marcucci, G., & Mazzuto, G. (2019). Fuzzy cognitive maps approach for analysing the domino effect of factors affecting supply chain resilience: A fashion industry case study. International Journal of Production Research, 1-29.

Ivanov, D., & Das, A. (2020). Coronavirus (COVID-19/SARS-CoV-2) and supply chain resilience: A research note. International Journal of Integrated Supply Management, 13(1), 90-102.

Ivanov, D., & Dolgui, A. (2019). Low-Certainty-Need (LCN) Supply Chains: A new perspective in managing disruption risks and resilience. International Journal of Production Research, 57(15-16), 5119-5136.

Ivanov, D., & Dolgui, A. (2020). Viability of intertwined supply networks: extending the supply chain resilience angles towards survivability. A position paper motivated by COVID-19 outbreak. International Journal of Production Research, 58(10), 2904-2915.

Ivanov, D., Dolgui, A., & Sokolov, B. (2019). The impact of digital technology and Industry 4.0 on the ripple effect and supply chain risk analytics. International Journal of Production Research, 57(3), 829-846.

Round 2

Reviewer 1 Report

I am satisfied with the Authors's response to my comments so I am recommending that this article is publishable.

Reviewer 2 Report

Thanks to the authors' effort to revise the script. 

Reviewer 3 Report

Dear Authors,

I read carefully your corrections and I hereby accept the paper in the present form.

Best Regards